# Wind Turbine Fault Detection Based on Time Series Monitoring Data

Qingyi Guo    Wang Xing    Linzhi Wang    Guohua Liu

## 1. Introduction

Wind energy has become a cornerstone of the global transition to renewable power. However, wind turbines often operate in harsh, remote environments, leading to high structural fatigue and frequent component failures. Traditional fault detection relies on analyzing isolated signals—such as vibration—in the time or frequency domains. While effective for localized issues, these methods struggle to distinguish between genuine faults and normal operational variability caused by fluctuating wind resources. This project proposes a **context-aware** framework that integrates heterogeneous data sources (temperature, electrical, and meteorological data) through advanced temporal networks, aiming to provide a more robust diagnostic tool.

### 1.1 Gaps Identified

**Information Silos:** Current research often treats different data sources as independent streams. This overlooks the physical correlation between the turbine's operational state and its mechanical response.

**Neglect of Environment Context:** Existing models typically monitor turbines in isolation. In reality, turbines are significantly subjected to fluctuations in external environmental factors, such as wind speed and atmospheric conditions. Ignoring environmental factors leads to high false-alarm rates when environmental turbulence is mistaken for mechanical anomalies.

**Static Feature Extraction:** Traditional frequency-domain analysis (e.g., FFT) assumes quasi-static conditions. However, wind turbines are highly non-stationary systems; existing methods often fail to capture the dynamic evolution of a fault as it progresses across different temporal scales.

### 1.2 Novelty

This project proposes a **context-aware Transformer-based anomaly detection framework** for wind turbine health monitoring. The key novelties are summarized as follows:

- **Context-aware Dependency Modeling:** The model captures complex relationships between *endogenous* and *exogenous* variables via an attention mechanism, enabling robust distinction between environmental variations and genuine fault patterns under nonstationary conditions.

- **Semantic-Enhanced Time-Series Representation:** Statistical descriptors (e.g., mean, variance, higher-order differences) are transformed into *textual semantic representations* and fused with raw time-series data, improving the detection of weak and distributed failure signals.

- **Unified Temporal-Semantic Fusion via Transformer:** A Transformer encoder jointly models temporal dependencies, cross-variable interactions, and semantic features within a unified framework, enhancing sensitivity to subtle early-stage anomalies.

## 2. Motivation

The maintenance of wind turbines, particularly offshore installations, accounts for up to 30% of the total levelized cost of electricity (LCOE). Unexpected gearbox or bearing failures can lead to prolonged downtime and substantial repair costs. Although modern turbines are equipped with abundant sensors, operators are frequently overwhelmed by nuisance alarms induced by the high variability of wind conditions. This underscores the need for diagnostic systems capable of reliably distinguishing transient environmental disturbances from incipient structural faults.

Despite the availability of rich sensor data, a fundamental theoretical gap persists in current health monitoring methodologies. Most existing approaches adopt a unimodal paradigm, treating Supervisory Control and Data Acquisition (SCADA) signals as independent streams and neglecting the intrinsic physical coupling within turbine systems. Moreover, traditional feature extraction techniques, such as frequency-domain analysis (e.g., FFT), rely on quasi-static assumptions and thus fail to capture the non-stationary and multi-scale temporal evolution of faults. As a result, these methods struggle to differentiate local faults from global environmental variations.

Integrating information across multiple physical domains offers the potential to enable context-aware diagnostics, facilitating the suppression of environmental noise and shifting fault detection from reactive signal matching toward proactive, system-level intelligence.

## 3. Existing Methods

Wind turbine fault detection is an application of **Prognostics and Health Management (PHM)**, framed primarily as a multivariate time-series anomaly detection or classification task. The goal is to identify deviations from a healthy state across major components like the gearbox, generator, and main bearing. Methods within this field can categorized into the following approaches:

### 3.1 Frequency-Domain Analysis

Traditional methods like Envelope Analysis (Cheng et al. 2018; Xin et al. 2012) and Spectral Kurtosis (Antoni 2006; Barszcz and Randall 2009) have been the gold standard for bearing fault detection. They excel at pinpointing specific mechanical components by identifying periodic shocks. However, these methods require constant rotational speeds to maintain frequency resolution. In variable-speed wind turbines, the frequency smearing effect often masks early-stage fault signatures, rendering these classical tools less effective without complex order-tracking compensation.

### 3.2 SCADA-based Performance Monitoring

Many studies utilize SCADA data to build "Normal Behavior Models" using Gaussian Process Regression (Wilkie and Galasso 2021; Avendano-Valencia, Chatzi, and Tcherniak 2020) or wavelet transformation (Tang, Liu, and Song 2010). These models predict the expected power output for a given wind speed; a significant residual between predicted and actual power indicates a fault. While excellent for detecting macroscopic issues (like pitch system failures), SCADA data lacks the sampling granularity to detect early-stage mechanical wear in high-speed components.

### 3.3 Deep Temporal Networks

The transition to Recurrent Neural Networks (RNNs) and Long Short-Term Memory (LSTM) units (Rama, Hur, and Yang 2024; Song et al. 2025) allowed researchers to model the temporal dependencies in sensor data. These models are far better at capturing the "evolution" of a fault over time compared to static snapshots. Nevertheless, standard LSTMs struggle with very long sequences and do not inherently account for relationships between sensors data within turbine and environmental factors.

## 4. Challenges

### 4.1 Weak and Distributed Failure Signals
Early failure signals are weak and spread across multiple telemetry channels. To detect these, the model must rely on features beyond raw data, such as semantic features derived from statistical measures like mean, variance, and higher-order differences. The challenge is effectively combining these derived features with raw time-series data to capture weak failure signals.

### 4.2 Complex dependencies between endogenous and exogenous variables
Capturing interdependencies between endogenous and exogenous variables is another challenge. Unlike traditional methods that analyze signals independently, the proposed method calculates attention matrices to capture variable relationships. Learning these relationships is difficult, especially with nonstationary and weak signals, but crucial for improving failure detection.

## 5. Objectives

The primary objective of this project is to develop a context-aware anomaly detection model for wind farm health monitoring based on context-aware model based on transformer architecture. The proposed model integrates information across heterogeneous physical domains and turbines monitoring data, enabling context-aware diagnostics. This design aims to enhance the model's ability to distinguish between environmental noise and genuine faults, thereby improving detection robustness. The effectiveness of the model will be systematically evaluated against existing wind turbine monitoring approaches, with ROC-AUC as the primary ranking metric.

# 6. Proposed Methodology

## 6.1 Architecture

The overall system follows an end-to-end pipeline that begins with raw wind turbine telemetry streams and ends with a binary early-warning score for failure. The system is based on a core Transformer encoder that computes attention matrices between endogenous and exogenous variables to capture the relationships among different attributes. In addition, semantic features such as the mean and variance of each attribute are then converted into text representations and concatenated with the time-series data, allowing the model to learn both temporal and semantic features.Our method enhances the model's ability to detect fault signals by combining attention matrices between variables and statistical features. Detailed specifications of the proposed model architecture are provided in Appendix B.

## 6.2 Data Collection and Preprocessing

The input data includes wind turbine telemetry, with attributes like bearing temperature, vibration, power output, wind speed, and ambient temperature. These metrics provide a comprehensive view of turbine conditions. During preprocessing, invalid data is removed, timestamps are ordered, and the data is segmented into fixed-length windows. A prediction-ahead strategy labels each window as positive if a failure occurs within the future horizon, and negative otherwise. Statistical features such as mean, variance, and higher-order differences are computed and used as semantic features to capture contextual information. A detailed description of the dataset is provided in Appendix A.

## 6.3 Evaluation Plan

The system will be evaluated using ROC-AUC as the primary ranking metric, alongside accuracy, precision, recall, and F1-score for performance at specific thresholds. It will be compared to baselines, including logistic regression and tree-based models. Ablation studies will assess the contributions of attention mechanisms, semantic features.

# 7. Progress So Far

**Completed**: A preliminary literature review has been conducted, covering wind turbine fault diagnosis and time series anomaly detection. In addition, the dataset has been collected, and an initial analysis of its key characteristics has been performed.

**Ongoing**: Current efforts are focused on processing the dataset to to align the data distribution with the model's expected input manifold, as well as expanding the literature review to establish a more comprehensive theoretical foundation for the proposed model design.

A detailed schedule of the project is provided in Appendix C.

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

## A. Dataset

This research uses a dataset collected from 70 wind turbines at a wind farm in China in 2025. The data includes SCADA records with a 1-minute sampling frequency, covering electrical signals (e.g., active power, voltage), mechanical signals (e.g., bearing temperature, vibration), and meteorological parameters (e.g., wind speed, ambient temperature). The data is stored in structured CSV files, representing a large-scale, multivariate time-series dataset for industrial monitoring.

Ground truth labels are derived from maintenance and fault logs, documenting downtime and component failures. These logs are aligned with the continuous operational data via timestamp matching, allowing each 1-minute entry to be mapped to a specific health state, distinguishing normal operation from failure modes.

Data preprocessing addresses issues like noise and gaps. Outliers and unphysical values are removed, missing values are filled using linear interpolation, and all features are normalized to a consistent scale to ensure proper learning from variables with different units (e.g., wind speed in m/s, power in kW).

## B. Core Model Design

The core prediction model is based on a Transformer encoder, which computes attention matrices between endogenous and exogenous variables to capture their relationships. This attention mechanism allows the model to identify dependencies between different monitoring attributes and environment factors, and understand how they contribute to potential failures. Additionally, semantic features are derived by calculating statistical properties like the mean and variance of each attribute, which are then converted into text representations. These representations are processed through a text encoder and concatenated with the time-series data, allowing the model to integrate both temporal and semantic information.

## C. Project Schedule

The proposed schedule of the project is shown in Table 1.

Table 1: Weekly Project Schedule

| Week | Dates | Tasks | Details | Lead/Support | |
|------|-------|-------|---------|--------------|---|
| 1 | May 4 - May 10 | Background study & dataset preprocessing | Review related work on wind turbine fault diagnosis, process the wind turbine dataset, and analyze data patterns | Wang(review), Linzhi(data) | |
| 2 | May 11 - May 17 | Core module development | Design and implement the anomaly detection model | Qingyi(model), Linzhi(support), Wang(support) | |
| 3 | May 18 - May 24 | Experiments implement | Training and testing the model, and conduct an ablation study | Qingyi(model), Linzhi(support), Wang(support) | |
| 4 | May 25 - May 31 | Finalization & report writing | Analyze results and compare with baselines, write the project report, and prepare presentation slides with a dry run | Linzhi(report), Qingyi(slide) | |

